# Splenic artery steal syndrome in patients with orthotopic liver transplant: Where to embolize the splenic artery?

**Florian N. Fleckenstein**[1,2]☯*, **Willie M. Luedemann**[1]☯, **Ahmet Kücükkaya**[1], **Timo A. Auer**[1,2], **Julius Plewe**[3], **Bernd Hamm**[1], **Rolf W. Günther**[1], **Uli Fehrenbach**[1], **Bernhard Gebauer**[1], **Gero Wieners**[1]

1 Department of Diagnostic and Interventional Radiology, Charité –Universitätsmedizin Berlin, Corporate Member of Freie Universität Berlin and Humboldt-Universität zu Berlin, Berlin, Germany, 2 Berlin Institute of Health at Charité –Universitätsmedizin Berlin, BIH Biomedical Innovation Academy, BIH Charité Clinician Scientist Program, Berlin, Germany, 3 Department of Abdominal Surgery, Charité –Universitätsmedizin Berlin, Corporate Member of Freie Universität Berlin and Humboldt-Universität zu Berlin, Berlin, Germany

☯ These authors contributed equally to this work.
* florian.fleckenstein@charite.de

## Abstract

### Purpose

This study compared proximal and distal embolization of the splenic artery (SA) in patients with splenic artery steal syndrome (SAS) after orthotopic liver transplantation (OLT) regarding post interventional changes of liver function to identify an ideal location of embolization.

### Methods and materials

85 patients with SAS after OLT treated with embolization of the SA between 2007 and 2017 were retrospectively reviewed. Periinterventional DSA was used to assess treatment success and to stratify patients according to the site of embolization. Liver function was assessed using following laboratory values: bilirubin, albumin, gamma-glutamyl transferase, glutamat-pyruvat-transaminase (GPT), glutamic-oxaloacetic transaminase (GOT), Alkaline Phosphatase (ALP), aPTT, prothrombin time and thrombocyte count. Descriptive statistics were used to summarize the data. Median laboratory values of pre, 1- and 3-days, as well as 1-week and 1-month post-embolization were compared between the respective embolization sites using linear mixed model regression analysis.

### Results

All procedures were technically successful and showed an improved blood flow in the hepatic artery post-embolization. Ten Patients were excluded due to re -intervention or inconsistent image documentation. Pairwise comparison using linear mixed model regression analysis showed a significant difference between proximal and distal embolization for GPT (57.0 (IQR 107.5) vs. 118.0 (IQR 254.0) U/l, p = 0.002) and GOT (48.0 (IQR 48.0) vs. 81.0 (IQR 115.0) U/l, p = 0.008) 3-days after embolization as well as median thrombocyte counts 7-days after embolization (122 (IQR 108) vs. 83 (IQR 74) in thousands, p = 0.014).

**Data Availability Statement:** All relevant data are within the paper and its Supporting Information files.

**Funding:** The author(s) received no specific funding for this work.

**Competing interests:** The authors have declared that no competing interests exist.

For all other laboratory values, no statistically significant difference could be shown with respect to the embolization site.

## Conclusion

We conclude that long-term outcomes after embolization of the SA in the scenario of SAS after OLT are irrespective of the site of embolization of the SA, whereas a proximal embolization potentially facilitates earlier normalization of liver function. Choice of technique should therefore be informed by anatomical conditions, safety considerations and preferences of the interventionalist.

## Introduction

For patients suffering from complex liver diseases such as advanced stages of cirrhosis, certain stages of liver cancer and congenital or acquired abnormalities in anatomy or metabolism of the liver, orthotopic liver transplantation (OLT) is usually the only curative treatment option. In this context, splenic artery steal syndrome (SAS) is a frequently described vascular complication after OLT with incidences reported between 3 and 6% in the literature [1–5]. SAS describes a pathological redirection of blood flow from the hepatic artery towards the splenic artery (SA) resulting in reduced hepatic arterial perfusion in the absence of an occlusion of the hepatic artery. This steal-situation might lead to ischemia, graft dysfunction and biliary damage, resulting in organ failure [2, 6, 7]. The clinical picture is non-specific which makes the diagnosis of SAS challenging [2, 8]. If clinically suspected, patients usually undergo a digital subtraction angiography (DSA) scan of the celiac axis. SAS is diagnosed by confirmation of an enlargement of the SA in combination with late enhancement of the hepatic artery territory [2, 9].

In this regard, embolization of the SA has been reported to be a safe and effective treatment to redirect blood flow to the hepatic circulation [2, 7, 10–12]. Better perfusion of the transplant graft after embolization is mainly due to two main effects. Firstly, it results in redirection of blood flow from the splenic flow territory to the hepatic artery territory increasing blood pressure in the common hepatic artery. Secondly, another mechanism has been discussed in the literature called hepatic artery buffer response (HABR) [13, 14]. It describes a reciprocal relationship between blood flow in the hepatic artery and the portal vein. A decrease of flow to the spleen automatically results in a lower blood volume in the portal vein. Through a negative feedback mechanism this finally leads to vasodilatation of hepatic arteries and an improved arterial blood flow to the liver. Embolization is commonly done using coils [10] or Amplatzer venous plugs [11], inserted into the SA via an intraarterial catheter until complete stasis is seen on DSA. However, there are risks and challenges associated with different approaches for embolotherapy whereas the ideal strategy is commonly informed by the target site. Data on the impact of the embolization site is scarce and a potentially ideal location in regard to graft function has not yet been evaluated [4, 11, 15, 16].

This study retrospectively investigated the possible superiority of either proximal or distal embolization of the SA in patients suffering from SAS after OLT regarding post-interventional changes of liver function.

## Methods and materials

### Patient selection

This retrospective cohort study was conducted in the radiological department of *blinded*. All patients provided written informed consent and the study was approved by the local ethics committee. Between November 2007 and January 2017, a total of 85 patients with SAS after OLT were referred to our department of interventional radiology for embolization of the SA. Inclusion criteria for this study were: (1) Embolization of the SA due to SAS, (2) Follow-up at least for 30 days after embolization and (3) no rejection of the transplant or major vascular complications during follow-up. A total of ten Patients had to be excluded after the initial search due to following reasons: Five patients had missing or incomplete image documentation which made it impossible to determine the exact location of embolization. Two patients had to undergo re-OLT due to transplant rejection within the first 30 days of OLT. Completely missing laboratory values lead to exclusion of two more patients. One patient received stenting of the hepatic artery in the same procedure in which embolization of the SA was performed because of a suspected partial dissection of the vessel. This patient was also excluded, leaving a total of 75 patients who were included into the final data analysis. Fig 1 shows the process of patient selection and -exclusion in a flowchart.

### Diagnosis of SAS

All diagnostics were based on the current understanding within the scientific community. Since SAS is a diagnosis of exclusion, patients were diagnosed in a stepwise approach [2, 7, 8, 17]. Initially, SAS was suspected due to elevated laboratory levels of transaminases, Alkaline Phosphatase (ALK) and Bilirubin or persistent ascites in the absence of acute cellular rejection, infection or toxicity.

Patients were then further examined with Doppler ultrasound. Dynamic assessment of velocity, waveforms, and resistance index (RI) of the abdominal arteries helps

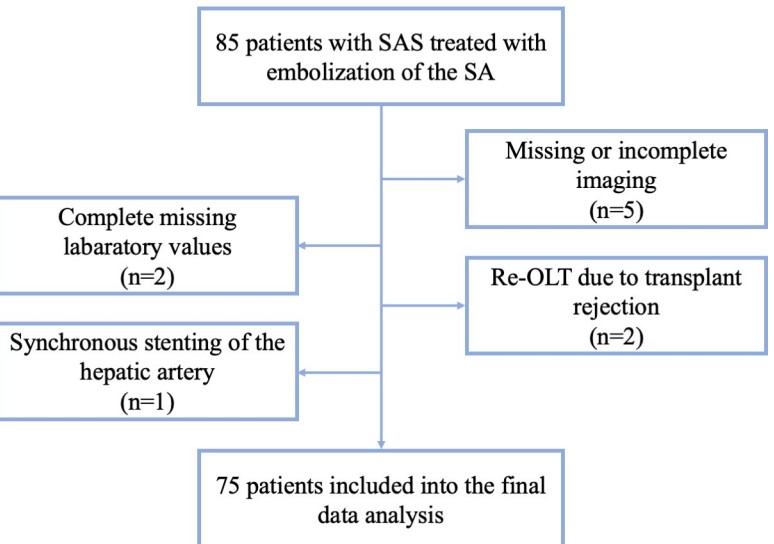

**Fig 1. Exclusion flowchart.** A total of ten patients were excluded. 75 patients were included in the final data analysis. Of these 75 patients a total of 60 patients had complete laboratory data during the 30-days follow up. 15 patients missing data but could be included into the linear mixed model regression analysis. SAS, Splenic Artery Steal Syndrome; SA, Splenic Artery; OLT, Orthotopic Liver Transplantation.

differentiate SAS from other causes of graft hypoperfusion such as hepatic artery stenosis or hepatic artery thrombosis. Patients suffering from SAS usually show high RIs (>0.8) in the hepatic artery with low diastolic flow or even reversal of diastolic flow. In addition, hepatic artery systolic velocities are unusually low (<35 cm/s) typically showing weak blood flow in the hepatic artery [17]. The diagnosis was confirmed if both, relative arterial hypoperfusion of the graft and an enlarged splenic artery was seen on DSA. Typical dynamic angiographic findings are (1) early perfusion of the splenic or gastroduodenal artery together with (2) delayed or dim perfusion of the hepatic artery along with (3) early portal venous contrast filling indicating hampered hepatic arterial blood flow (Fig 2A–2C) [7].

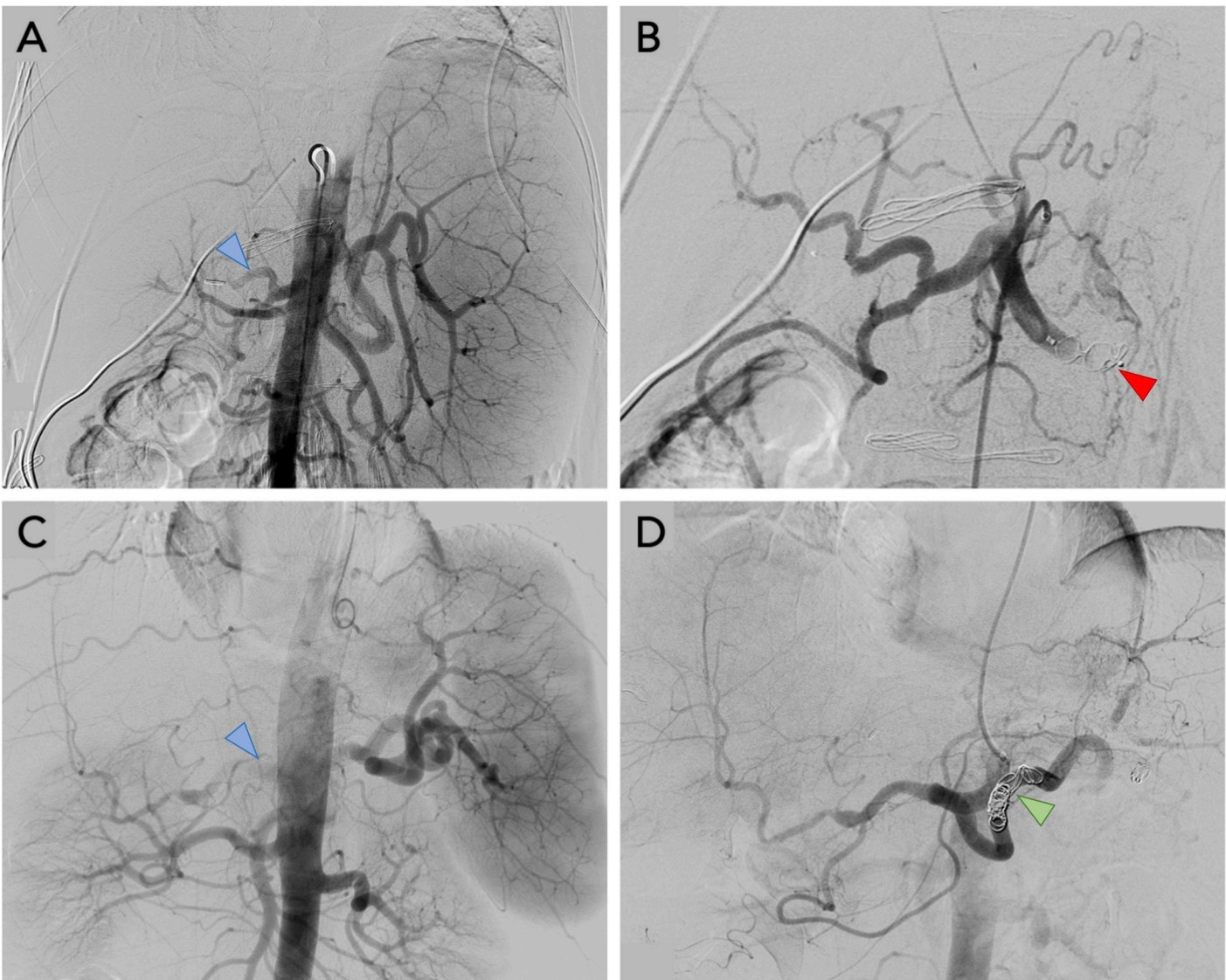

**Fig 2. Abdominal DSA.** A-C. Pre-intervention celiac angiography shows hampered hepatic arterial flow with delayed contrast filling of the hepatic artery (blue arrows) as well as early and strong contrast filling of the enlarged splenic artery. B-D. Post-interventional celiac angiography shows significantly improved hepatic arterial flow and delayed contrast filling of the splenic artery. B. shows embolization of the SA using an Amplatzer venous plug (red marker), D. shows an embolization using coils (green marker).

## Procedure techniques

Board-certified radiologists all with > 10 years of experience, performed the interventions. After local anesthesia with lidocaine 1%, vascular access in micropuncture technique was performed preferably in the right common femoral artery or left brachial artery and a 6F vascular sheath was placed in Seldinger-technique via a 0.035-inch guide wire. An overview angiography of the celiac trunk and the superior mesenteric artery was generated with a 5-F Cobra (Radifocus, Terumo Europe NV) or a 5-F SOS Omni Selective catheter (Soft-Vu, Angiodynamics) before advancement into the splenic artery (Fig 2A–2C). Additional DSA runs allowed for analysis of splenic artery anatomy and size. The catheter was advanced into a favorable position. If advancement of the macrocatheter was not possible, the SA was accessed via a microcatheter (Cantata 2.5 F or MicroFerret-18 3 F, Cook Medical, Bloomington, IN). For embolization with either an Amplatzer plug (AGA Medical, Golden Valley, MN) a 6F curved sheath was introduced into the SA (Fig 2B). Coil embolization was either done using the macro- or microcatheter (Fig 2D). Gianturco coils of variable of various sizes (Cook Medical, Bloomington, IN) were used. Embolization was performed until stasis of blood flow was seen on DSA (Fig 2B–2D). Following OLT, all patients received a calcineurin-inhibitor-based immunosuppressive protocol and steroids which are usually tapered within the first month after surgery. NSAR or antibiotics are not part of our standard of care for SA embolization but may have been administered at discretion of the referring department depending on the clinical scenario.

Periinterventional DSA was retrospectively used to assess treatment success and to stratify patients according to the site of embolization. Liver function was assessed using following laboratory values: bilirubin (TBIL), albumin, gamma-glutamyl transferase (GGT), glutamat-pyruvat-transaminase (GPT), glutamic-oxaloacetic transaminase (GOT), ALP, aPTT, prothrombin time (PT) using international normalized ratio (INR) and thrombocyte count.

## Statistical analysis

Descriptive statistics were used to summarize the data in absolute numbers and percent. Testing for normality was performed with the Shapiro Wilk test. Non-normally distributed data are expressed as median and interquartile range (IQR) and compared with the non-parametric Mann-Whitney U test and Wilcoxon signed-rank test. Median laboratory values of pre, one-, 3-days, as well as 1-week and 1-month post embolization were compared. Linear mixed model regression analysis was used to identify significant effects of the embolization site on repeated measurements of the above-mentioned laboratory values across different time points. Conception of the longitudinal mixed model was informed by the impact on the Akaike's and Bayesian information criteria which both penalize for the complexity of a given model in order to minimize overfitting when explaining the variance in the data. For the model used in this study, time was treated as a categorial variable. The models included the embolization site, time, baseline scores, age as well as embolization site * time interactions as fixed covariates. The intercept was modeled as random effect. We expected the correlation between measurements to decrease with increased time between the measurements which is why an autoregressive co-variance structure was chosen. The results were normalized through transformation with the natural logarithm to allow for pairwise comparison with the standardized mean difference (SMD) and visualization with Forest plots. All statistical analyses were performed using IBM SPSS STATISTICS, version 25 (IBM Corporation, Armonk, NY, USA). Hypothesis tests were 2-tailed and used a 5% significance level.

# Results

Median patient age of the patient cohort was 52.8 (IQR 13) years. Complete documentation of laboratory values after 30 days was available for 60 patients, incomplete data was available for a total of 75 patients. After 30 days, all included patients were alive. Table 1 summarizes patient characteristics of our cohort. In most patients (81%, n = 61), a femoral access was used, a brachial access was used in only 19% (n = 14) of cases. Embolization was done using Amplatzer venous plugs in 76% (n = 57) and coils in 11% (n = 7) of cases. In 13% (n = 11) a combination of both was used. A proximal site of embolization of the SA was chosen in 55% (n = 41) of cases. All procedures were technically successful resulting in an improved blood flow in the common hepatic artery post-embolization by means of angiographic criteria [7]. Of note, none of the patients suffered from a procedure related complication such as infections or relevant peri-interventional pain.

Using the Wilcoxon signed-rank-test for the comparison of dependent measurements at the beginning and the end of the observational period for 63 patients with complete data, a significant decrease of median laboratory values over the course of 30 days could be detected for TBIL by 50.2% (2.95 (IQR 6.94) to 1.47 (IQR 3.06) mg/l), p = 0.003, 78% of patients showing normalized values), GPT by 47.5% (80 (IQR 208) to 42 (IQR 39) U/l), p<0.001, 83% of patients showing normalized values), GOT by 47.7% (66 (IQR 365.2) to 34.5 (IQR 48.5) U/l), p<0.001, 42% of patients showing normalized values), and PT by 4.4% (1.26 (IQR 0.33) to 1.21 (IQR 0.28), p = 0.013, 72% of patients showing normalized values). A significant increase could be shown for albumin by 4.3% (30.1 (IQR 6.2) to 31.3 (IQR 7.7) g/l, p = 0.002, 61% showing normalized values), for ALP by 66.1% (121 (IQR 121) to 201 (IQR 231) U/l), p = 0.001, 47%

**Table 1. Patient characteristics.**

| Parameter | | N (%) |
|---|---|---|
| *Demographics* | | |
| Number of patients | | 75 (100) |
| Age | Median (IQR) | 52.8 (13) |
| | >60 years | 25 (30) |
| | <60 years | 50 (70) |
| Sex (female) (%) | | 27/75 (36%) |
| Indication for OLT | Alcohol liver cirrhosis | 28 (37) |
| | Cryptogene cirrhosis | 20 (27) |
| | HCC | 17 (23) |
| | HCV cirrhosis | 9 (12) |
| | Primary biliary cirrhosis | 1 (1) |
| *Treatment* | | |
| Site of embolization | proximal | 41 (55) |
| | distal | 34 (45) |
| Access | femoral | 61 (81) |
| | radial | 14 (19) |
| Embolization Material | Coils | 7 (11) |
| | Amplatzer Plaque | 57 (76) |
| | Coils + Plaques | 11 (13) |
| Median time from OLT to treatment in days (IQR) | | 44 (53.5) |

SD, standard deviation; OLT, orthotopic liver transplantation; HCC, hepatocellular carcinoma; HCV, hepatitis C virus.

showing normalized values) and for thrombocytes count by 62.5% (80T (IQR 127) to 130T (IQR 100.5), p<0.001, 66% showing normalized values after 30 days), respectively. Changes of laboratory values were non-significant after 30 days for GGT (+9.1% (175 (IQR 225) to 159 (IQR 229) U/l), p = 0.341) and aPTT (-10.7% (36.8s (IQR 8.2) to 40.7s (IQR 13.6), p = 0.07, 33% of patients showing normalized values). Fig 3 shows the development of median laboratory values over time stratified regarding the site of embolization.

According to the Mann-Whitney U test, there were no statistically significant differences of median laboratory values between the cohorts of proximal and distal location of embolization at any time point. Notably, no significant difference could be detected between both patient groups at baseline. The results of the longitudinal mixed model analyses along with forest plots are reported in Fig 4. Pairwise comparison revealed a significant difference between proximal and distal embolization for GPT (57.0 (IQR 107.5) vs. 118.0 (IQR 254.0) U/l, p = 0.002) and GOT (48.0 (IQR 48.0) vs. 81.0 (IQR 115.0) U/l, p = 0.008) 3-days after embolization. Furthermore, a significant difference could be detected for median thrombocyte counts 7-days after embolization (122 (IQR 108) vs. 83 (IQR 74) in thousands, p = 0.014). For all other laboratory values, no statistically significant difference could be shown between both groups stratified regarding the site of embolization.

## Discussion

This study could show that a successful embolization of SAS after OLT is independent of the localization of embolization of the SA.

In detail our results draw two main findings: Firstly, as evaluated by laboratory values, embolization of the SA is an effective treatment of SAS to improve function of the OLT. Secondly, although we could encounter statistically significant differences between proximal and distal embolization for GPT, GOT and for thrombocyte counts at single time points, for the vast majority of laboratory values and time points no statistically significant difference could be shown. Moreover, laboratory values tend to match after 30 days indicating a self-adjusting equilibrium.

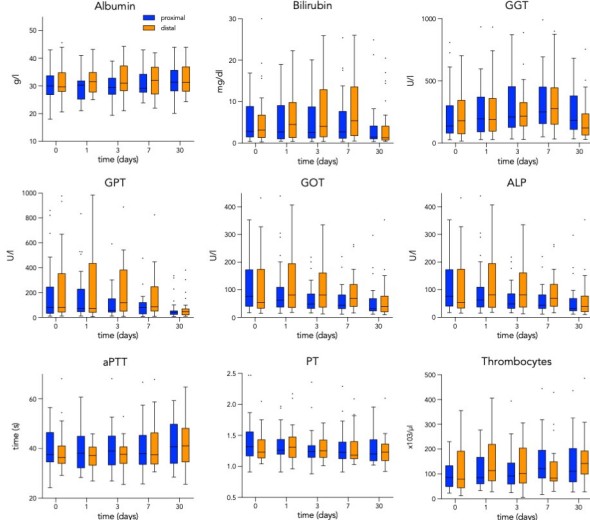

**Fig 3. Boxplots showing the development of median laboratory values over time stratified regarding the site of embolization.** GGT, gamma-glutamyl transferase, GPT, glutamic-pyruvic transaminase; GOT, glutamic-oxaloacetic transaminase; ALP, Alkaline Phosphatase; PT, Prothrombin Time using International Normalized Ratio (INR).

| Parameters of liver function | Median values (IQR) | | Pairwise comparison between Group Difference (95% CI) | SMD analysis | |
|---|---|---|---|---|---|
| | Proximal embolization | Distal embolization | | Proximal embolization | Distal embolization |
| **After 1 day** | | | | | |
| Albumin (g/l) | 30.3 (6.3) | 31.5 (6.5) | -1.8 (-4.1 – 0.6) | | |
| Bilirubin (mg/dl) | 2.7 (7.2) | 4.5 (7.5) | -1.1 (-3.1 – 1.0) | | |
| GGT (U/l) | 195.5 (246.5) | 188.0 (248.0) | 3.5 (-122.4 – 129.4) | | |
| GPT (U/l) | 74.0 (174.0) | 74.0 (295.8) | 11.8 (-105.6 – 129.3) | | |
| GOT (U/l) | 62.5 (65.8) | 81.8 (115.0) | -42.4 (-147.9 – 63.0) | | |
| ALP (U/l) | 123.0 (150.5) | 15.0 (111.8) | -24.2 (-110.0 – 61.5) | | |
| aPTT (s) | 81.1 (11.9) | 37.2 (7.1) | -3.4 (-8.2 – 1.5) | | |
| PT | 1.3 (0.2) | 1.3 (0.3) | -0.0 (-0.1 – 0.2) | | |
| Thrombocytes | 85 (100) | 114 (143) | -13.1 (-46.4 – 20.1) | | |
| | | | | | |
| **After 3 days** | | | | | |
| Albumin (g/l) | 29.4 (5.6) | 31.0 (8.3) | -2.3 (-4.6 – 0.0) | | |
| Bilirubin (mg/dl) | 2.6 (7.3) | 4.0 (7.8) | -1.5 (-3.6 – 0.7) | | |
| GGT (U/l) | 210.5 (323.3) | 216.0 (141.0) | 56.5 (-71.7 – 184.8) | | |
| GPT (U/l) | 57.0 (107.5) | 118.0 (254.0) | -189.2* (-308.7 – -69.6) | | |
| GOT (U/l) | 48.0 (48.0) | 81.0 (115.0) | -147.4* (-255.7 – -39.0) | | |
| ALP (U/l) | 130.0 (146.0) | 150.0 (105.0) | -43.6 (-130.2 – 43.0) | | |
| aPTT (s) | 38.4 (10.2) | 37.6 (6.6) | 0.9 (-3.9 – 5.8) | | |
| PT | 1.2 (0.2) | 1.3 (0.2) | 0.1 (-0.1 – 0.2) | | |
| Thrombocytes | 92 (79) | 104 (139) | 9.2 (-25.3 – 43.7) | | |
| | | | | | |
| **After 7 days** | | | | | |
| Albumin (g/l) | 29.2 (6.1) | 32.0 (7.0) | -1.9 (-4.3 – 0.5) | | |
| Bilirubin (mg/dl) | 2.7 (6.3) | 5.4 (11.2) | 0.2 (-1.9 – 2.4) | | |
| GGT (U/l) | 251.0 (276.0) | 276.0 (180.0) | -28.7 (-161.0 – 103.4) | | |
| GPT (U/l) | 78.0 (90.5) | 86.5 (175.8) | -36.0 (-160.0 – 87.9) | | |
| GOT (U/l) | 44.0 (44.5) | 68.5 (72.3) | -34.8 (-147.1 – 77.5) | | |
| ALP (U/l) | 191.0 (149.3) | 175.0 (196.0) | 5.9 (-83.1 – 94.9) | | |
| aPTT (s) | 37.9 (11.0) | 37.3 (8.6) | -0.4 (-5.5 – 4.7) | | |
| PT | 1.2 (0.3) | 1.2 (0.3) | 0.0 (-0.1 – 0.2) | | |
| Thrombocytes | 122 (108) | 83 (74) | 44.2* (9.0 – 79.4) | | |
| | | | | | |
| **After 30 days** | | | | | |
| Albumin (g/l) | 31.4 (7.1) | 31.3 (8.8) | 0.5 (-1.8 – 2.9) | | |
| Bilirubin (mg/dl) | 1.5 (2.8) | 1.3 (2.9) | 1.8 (-0.4 – 3.9) | | |
| GGT (U/l) | 184.0 (269.0) | 122.0 (162.0) | -24.5 (-155.2 – 106.2) | | |
| GPT (U/l) | 41.0 (30.8) | 46.0 (46.0) | -14.6 (-136.7 – 107.4) | | |
| GOT (U/l) | 32.0 (45.0) | 40.0 (58.0) | -3.9 (-115.6 – 107.9) | | |
| ALP (U/l) | 218.0 (210.0) | 167.0 (257.0) | 0.4 (-87.5 – 88.3) | | |
| aPTT (s) | 40.4 (14.3) | 41.1 (13.2) | -0.5 (-5.5 – 4.4) | | |
| PT | 1.2 (0.3) | 1.2 (0.3) | 0.0 (-0.1 – 0.2) | | |
| Thrombocytes | 111 (125) | 142 (93) | 7.2 (-28.1 – 42.5) | | |

**Fig 4. Median values as well as pairwise comparisons differentiated after location of embolization of the SA.** * The mean difference is significant at the .05 level. IQR, interquartile range; SA, splenic artery; CI, confidence interval; GGT, gamma-glutamyl transferase, GPT, glutamic-pyruvic transaminase; GOT, glutamic-oxaloacetic transaminase; ALP, Alkaline Phosphatase; PT, Prothrombin Time. Forest plot analysis of the log-transformed standardized mean difference (SMD) for pairwise comparison.

Coil and Amplatzer venous plug embolizations are well described in the literature and have been successfully implemented in daily clinical routine [4, 7, 11, 18]. However, several limitations have been reported for positioning of embolic material with multiple cases of dislocation and splenic infarction as a result of far distal embolization [4, 16, 19].

Besides operator preference for embolic materials the question of where to embolize is not easy to answer. As stated in the literature, more proximal occlusion of the SA lowers the risk of infarction of the spleen by perfusion through arteries from the gastric, pancreatic, and

gastroepiploic territory, hence preserving a sufficient perfusion of the spleen [15]. Our data shows, that a more proximal embolization is associated with significantly higher thrombocyte counts 3-days post intervention. This might be due to an overall better perfusion of the spleen and hence, a mostly preserved platelet physiology [20]. The statistically significant, transient increase in GPT and GOT and the not significant increase in ALP and bilirubin levels after distal embolization appear to be in certain contrast to the more consistent down-trending of the aforementioned parameters after proximal embolization. This might indicate a transient or rather prolonged liver hypoperfusion after distal embolization but a similar long-term outcome, presumably due to autoregulation.

Especially in situations of daily clinical routine an embolization of proximal parts of the SA might be more feasible and faster without the use on microcatheters [11] and allows for the use of both, Amplatzer venous plugs and coils as used in parts of our study cohort [7, 11, 16]. Our results encourage interventionalists to be free to use whatever material they are most familiar with. We believe this might ultimately result in higher success rates and better patient outcomes. To our knowledge, this is the first study that assessed the impact of the embolization site with respect to changes of liver-related laboratory values over time.

This study has several limitations. It is a single-institutional retrospective analysis of 75 patients with SAS after OLT treated with embolization of the SA. The medium-sized cohort with fairly heterogeneous underlying diseases lacks subgroup analysis and might therefore not be representative. Being one of the largest transplant centers in Germany however, our study includes all patients that were treated in our institution and it will be challenging to find a larger cohort without endangering the consistency of the study protocol. Due to inconsistent follow-up imaging a volumetric assessment of the size of the spleen was not possible. At this point we're conducting further studies to evaluate changes of spleen volume after embolization in order to predict successful treatment of SAS.

## Conclusion

This study could show that long-term outcomes after embolization of the SA in the scenario of SAS after OLT are independent of the localization of embolization of the SA, whereas a proximal embolization potentially facilitates earlier normalization of liver function. Choice of embolization technique and site might therefore solely be based on interventionalists preferences and anatomical conditions.

## Supporting information

**S1 Data.**
(XLSX)

## Author Contributions

**Conceptualization:** Florian N. Fleckenstein, Gero Wieners.

**Data curation:** Florian N. Fleckenstein, Willie M. Luedemann, Ahmet Kücükkaya.

**Formal analysis:** Florian N. Fleckenstein, Timo A. Auer.

**Investigation:** Florian N. Fleckenstein.

**Methodology:** Florian N. Fleckenstein, Gero Wieners.

**Project administration:** Florian N. Fleckenstein, Gero Wieners.

**Supervision:** Willie M. Luedemann, Bernd Hamm, Rolf W. Günther, Uli Fehrenbach, Bernhard Gebauer, Gero Wieners.

**Validation:** Florian N. Fleckenstein, Willie M. Luedemann, Uli Fehrenbach.

**Writing – original draft:** Florian N. Fleckenstein.

**Writing – review & editing:** Florian N. Fleckenstein, Willie M. Luedemann, Julius Plewe, Uli Fehrenbach, Gero Wieners.

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
