## [Decision Letter · Decision Letter 0]

20 Sep 2021

PONE-D-21-24840Splenic artery steal syndrome in patients with orthotopic liver transplant: where to embolize the splenic artery?PLOS ONE

Dear Dr. Fleckenstein,

Thank you for submitting your manuscript to PLOS ONE. After careful consideration, we feel that it has merit but does not fully meet PLOS ONE’s publication criteria as it currently stands. Therefore, we invite you to submit a revised version of the manuscript that addresses the points raised during the review process. As you can see, both reviewers evaluated your paper as potentially interesting, but requested several important additions that are needed before the paper can be evaluated further.

We look forward to receiving your revised manuscript.

Kind regards,

Pavel Strnad

Academic Editor

PLOS ONE

Journal Requirements:

Note: HTML markup is below. Please do not edit.]

Reviewers' comments:

Reviewer's Responses to Questions

**Comments to the Author**

1. Is the manuscript technically sound, and do the data support the conclusions?

Reviewer #1: Partly

Reviewer #2: Yes

2. Has the statistical analysis been performed appropriately and rigorously? 

Reviewer #1: No

Reviewer #2: I Don't Know

3. Have the authors made all data underlying the findings in their manuscript fully available?

Reviewer #1: No

Reviewer #2: Yes

4. Is the manuscript presented in an intelligible fashion and written in standard English?

Reviewer #1: No

Reviewer #2: Yes

5. Review Comments to the Author

Reviewer #1: The authors presented a retrospective evaluation of the splenic artery embolization in patients with steal syndrome after orthotopic liver transplantation. They compared the effect of proximal and distal embolization and they concluded that the site of the embolization had no impact on the beneficial effect of the intervention. The conclusion is based only on the comparison of the course of laboratory parameters during the 30 days after the procedure. The paper might bring new interesting facts but the data should be better described and interpreted. The manuscript needs a major revision to be suitable for publication.

Major comments

1. The patient's selection process must be better explained. There were 85 patients who underwent SA embolization for SASS but 10 of them were excluded from further evaluation because of the need of re-intervention which was not related to the SA embolization. However, the term re-intervention implicates the sensation that the patients needed the same procedure again. This point should be explained in detail. Were there patients with inefficient first embolization procedure? If yes, how were they distributed between the proximal and distal embolization site subgroups?

2. The laboratory data during the observation period were available only in 60 patients. The numbers of the evaluated patients must be added in the appropriate columns in the table 2. A flow chart showing the number of included, excluded and evaluated patients might be helpful for better understanding the results.

3. The process leading to the SAS diagnosis is unclear. The authors wrote that patients had had an alteration in specific liver laboratory values. When looking at figure 2, it seems that there were also patients with normal laboratory values at day 0. What was the role of ultrasound in the decision to perform DSA? Were there also patients with weak hepatic artery blood flow on ultrasound but with normal laboratory parameters?

4. The authors should better document the improvement of liver graft function after the SA embolization in each subgroup of patients. They should add the proportions of patients who normalised GPT and other enzymes or achieved normal platelet count after the 30-days observation period.

Minor comments

Some abbreviations were used incorrectly in the manuscript:

1. The authors defined only abbreviation SA for splenic artery, but they used also SAE for splenic artery embolization. SAE represents a generally accepted abbreviation for serious adverse event. I would recommend not to use this abbreviation for a different term.

2. GOT and AST are two abbreviations for the same enzyme (Aspartate transaminase (AST) or aspartate aminotransferase, also known as AspAT/ASAT/AAT or glutamic oxaloacetic transaminase (GOT, SGOT), is a pyridoxal phosphate (PLP)-dependent transaminase enzyme (EC 2.6.1.1) but they were presented in the text and also in the table 2 and figure 2 as two different enzymes.

3. AP represents the abbreviation for the enzyme alkaline phosphatase (EC 3.1.3.1). The values of AP were presented in the section Results, but it was not mentioned in the section Procedure techniques. Furthermore, the abbreviation AP is missing in the comment under the table 2 and figure 2.

4. INR (international normalized ratio) is a standardised method for the expression of prothrombin time results. The assessed parameter should be called prothrombin time (PT) and INR as the mode of measurement (unit).

5. The authors used SAS and also SASS without explaining the difference between both abbreviations.

Reviewer #2: In a retrospective cohort study, the authors compared proximal versus distal splenic artery embolization in patients with splenic artery steal syndrome after liver transplantation with respect to postinterventional changes in surrogate markers of liver function. This study is of importance because the optimal site for embolization is unclear and few data are available on this topic. They observed a median increase in GPT levels after distal embolization but a decrease after proximal embolization and improvement in platelet counts at 3 and 7 days. The manuscript is well written and the data are well presented.

Key Points:

Methods: Please explain in more detail the use of the mixed linear regression model (fixed and random effects). Why do the authors believe this model is superior to the nonparametric comparison of laboratory values at time 0 and time x using the signed-rank test?

Figure 2: To better assess the change in transaminase levels after proximal versus distal embolization, I propose to plot median delta transaminase levels (compared with day 0) for all time points.

Do the authors have data on white blood cell counts or CRP levels after embolization to assess whether systemic inflammation differs between proximal and distal embolization?

The increase in GPT levels at 3 days alongside similar levels at 30 days could indicate transient liver hypoperfusion after the procedure but a similar long-term outcome. Please discuss.

Minor points:

The authors state that there were no procedure-related complications. Does this apply to pain, splenic abscesses, and infections?

Were patients routinely treated with nonsteroidal anti-inflammatory drugs, steroids, or antibiotic prophylaxis?

Please provide any other characteristics of the study cohort, including time since transplant and immunosuppression, that might affect transaminase progression.

Was hepatic artery stenosis excluded in all patients?

6. PLOS authors have the option to publish the peer review history of their article (what does this mean?). If published, this will include your full peer review and any attached files.

Reviewer #1: No

Reviewer #2: No

---

## [Author Response · Author response to Decision Letter 0]

3 Jan 2022

5. Review Comments to the Author

Reviewer #1: The authors presented a retrospective evaluation of the splenic artery embolization in patients with steal syndrome after orthotopic liver transplantation. They compared the effect of proximal and distal embolization and they concluded that the site of the embolization had no impact on the beneficial effect of the intervention. The conclusion is based only on the comparison of the course of laboratory parameters during the 30 days after the procedure. The paper might bring new interesting facts but the data should be better described and interpreted. The manuscript needs a major revision to be suitable for publication.

Major comments

1. The patient's selection process must be better explained. There were 85 patients who underwent SA embolization for SASS but 10 of them were excluded from further evaluation because of the need of re-intervention which was not related to the SA embolization. However, the term re-intervention implicates the sensation that the patients needed the same procedure again. This point should be explained in detail. Were there patients with inefficient first embolization procedure? If yes, how were they distributed between the proximal and distal embolization site subgroups?

We fully agree with the reviewer and added a paragraph regarding the exclusion of patients during the selection process. Five patients were excluded due to missing/incomplete imaging, two because of missing lab values as well as re-OLT after organ rejection. One patient received a stent placement in the hepatic artery synchronous with the embolization procedure. Indication for this was a partial dissection of the hepatic artery. Please see also figure 1 where we clarified the selection process in a flow chart diagram as suggested in comment 2.

2. The laboratory data during the observation period were available only in 60 patients. The numbers of the evaluated patients must be added in the appropriate columns in the table 2. A flow chart showing the number of included, excluded and evaluated patients might be helpful for better understanding the results.

Thank you very much for this comment. 60 patients had all laboratory data available at all time points. Out of 75 patients a total of 15 patients had at least one laboratory value missing. Yet, linear mixed model regression analysis allowed us to include patients with missing values into the analysis to strengthen our data. In fact, this was the reason why we chose this statistical model over a simpler repeated measures ANOVA approach. Hence, we were able to include all 75 patients into the statistical analysis. We clarified this in the methods and results section in order to explain this to potential readers. Also thank you very much for the suggestion of a flow chart. Please see figure 1. We believe this figure clarifies also parts of comment 1. 

3. The process leading to the SAS diagnosis is unclear. The authors wrote that patients had had an alteration in specific liver laboratory values. When looking at figure 2, it seems that there were also patients with normal laboratory values at day 0. What was the role of ultrasound in the decision to perform DSA? Were there also patients with weak hepatic artery blood flow on ultrasound but with normal laboratory parameters?

This is a very helpful comment by the reviewer,. SAS is a diagnosis of exclusion and although conventional angiography is necessary in the diagnosis of SAS, no true gold standard for imaging exists. The diagnosis is suspected based upon a constellation of clinical, laboratory, and imaging findings after exclusion of more common causes of graft dysfunction. Ultimately, the diagnosis of SAS is made ex juvantibus when increased hepatic arterial perfusion and improved graft function are seen after splenic artery embolization.

Hence, we use a stepwise approach to narrow down the differential diagnosis before any intervention. After exclusion of cellular rejection, infection or toxicity, Doppler ultrasound is used to rule out vascular and biliary complications in OLT patients. Evaluation of hepatic artery velocity, waveforms, and particularly vascular resistance helps differentiate SAS from Hepatic Artery Thrombosis and Hepatic Artery Stenosis with collateralization. However, most findings are non-specific and could be caused by transient graft edema, organ rejection, or infection. 

Like assumed by the reviewer, there were indeed patients showing normal laboratory values at baseline. Yet, even if some laboratory values were normal, they were always accompanied by abnormalities of other surrogate parameters of liver function. Besides, in some patients Doppler Ultrasound was also performed if persistent ascites was seen in patients without any other cause for this such as rejection or infection. We clarified our diagnostic workflow in the methods part of the manuscript with emphasis on the stepwise approach including the role of Doppler Ultrasound. 

4. The authors should better document the improvement of liver graft function after the SA embolization in each subgroup of patients. They should add the proportions of patients who normalized GPT and other enzymes or achieved normal platelet count after the 30-days observation period.

Thank you for this comment. We added the percentage of normalized laboratory values after 30 days in the results section for every value evaluated. Although all procedures were technically successful, we were not able to achieve an absolute normalization in all values. Yet, when looking at the statistical analysis we could achieve a significant improvement of all laboratory values but GGT in the 30-days post-intervention. We clarified this in the results section of the manuscript.

Minor comments

Some abbreviations were used incorrectly in the manuscript:

1. The authors defined only abbreviation SA for splenic artery, but they used also SAE for splenic artery embolization. SAE represents a generally accepted abbreviation for serious adverse event. I would recommend not to use this abbreviation for a different term.

We standardized our abbreviations accordingly throughout the manuscript. For example, “embolization of the SA” is simply used instead of SAE.

2. GOT and AST are two abbreviations for the same enzyme (Aspartate transaminase (AST) or aspartate aminotransferase, also known as AspAT/ASAT/AAT or glutamic oxaloacetic transaminase (GOT, SGOT), is a pyridoxal phosphate (PLP)-dependent transaminase enzyme (EC 2.6.1.1) but they were presented in the text and also in the table 2 and figure 2 as two different enzymes.

Thank you for this comment. We corrected double use of the two names for the same enzyme in all sections of the manuscript including abbreviation section of figures and tables.

3. AP represents the abbreviation for the enzyme alkaline phosphatase (EC 3.1.3.1). The values of AP were presented in the section Results, but it was not mentioned in the section Procedure techniques. Furthermore, the abbreviation AP is missing in the comment under the table 2 and figure 2.

We added alkaline phosphatase in the methods and material part of the manuscript as well as in table 2 and figure 3. We furthermore changed the abbreviation AP to ALP which is the more commonly used term in English literature. Thank you very much!

4. INR (international normalized ratio) is a standardised method for the expression of prothrombin time results. The assessed parameter should be called prothrombin time (PT) and INR as the mode of measurement (unit).

We implemented prothrombin time for the assessed parameter in the manuscript as well as in all tables and figures

5. The authors used SAS and also SASS without explaining the difference between both abbreviations.

There is no difference, we corrected the abbreviation and standardized using the term SAS for “splenic artery steal syndrome”

Reviewer #2: In a retrospective cohort study, the authors compared proximal versus distal splenic artery embolization in patients with splenic artery steal syndrome after liver transplantation with respect to postinterventional changes in surrogate markers of liver function. This study is of importance because the optimal site for embolization is unclear and few data are available on this topic. They observed a median increase in GPT levels after distal embolization but a decrease after proximal embolization and improvement in platelet counts at 3 and 7 days. The manuscript is well written and the data are well presented.

Key Points:

Methods: Please explain in more detail the use of the mixed linear regression model (fixed and random effects). Why do the authors believe this model is superior to the nonparametric comparison of laboratory values at time 0 and time x using the signed-rank test?

Treatment outcomes can be analyzed using only a final measurement like a Wilcoxon signed-rank test. 

Unfortunately, much of the information captured with repeated measurements, such as the pattern of outcomes across the timepoints, is not assessed. Another weakness of the signed-rank test is its limited power. As observations are converted to ranks and only ranks are used in the test statistic, the signed-rank test does not use all available information in the original data, leading to lower power when compared with tests that use all data. Furthermore, missing values in the data set lead to the exclusion of the pair. Nonetheless, for descriptive data, we used the signed rank test to compare median baseline values and values at the end of the observation period of 30 days. We deemed it best to provide an outline of the results with a test that is familiar to most readers and more comprehensible than the models discussed below. 

Linear Mixed Models (LMM) account for correlations between repeated measurements within each patient and are well suited to settings in which the individual trajectory of a particular outcome for a study participant over time is influenced both by factors that can be assumed to be the same across a collective, so-called fixed effects (such as the effect of an intervention), and by factors that are likely to vary from patient to patient, so-called random effects. The results reported after 30 days show the impact of the intervention on the trajectory of the patient across the different time points and do not only compare the first and the last data point as e.g. a signed-rank test. Repeated measures analysis of variance (rmANOVA) e.g., which is often used for analyzing longitudinal data, does not have the same flexibility as LMM and can yield misleading results due to its more rigid assumptions (e.g. all effects are considered fixed). Another advantage of mixed modeling over rmANOVAS is the handling of missing data. Under the “missing-at-random” assumption, mixed models can accommodate unbalanced data patterns and use all available data points and patients in the analysis, resulting in a more powerful study. 

To conclude, we believed the advantages of mixed models over e.g. a signed rank test and rmANOVAs to be the additional information inferred from the trajectory of an individual patient across different time points, the flexibility of model building in terms of fixed and random effects and the superior manageability of missing data and statistical power. 

We totally agree with you that a signed-rank test and LMM, under certain circumstances, shed light on the same question from different angles which might be confusing for readers. We went into greater detail of our model building approach in the methods section and restructured the results section accordingly. If you still find the use of to different test statistics confusing for potential readers, we would report solely the results of our longitudinal mixed model analyses.

Figure 2: To better assess the change in transaminase levels after proximal versus distal embolization, I propose to plot median delta transaminase levels (compared with day 0) for all time points.

We felt the reporting of median delta values might cause confusion as e.g. bilirubin, thrombocytes and transaminases distribute across entirely different ranges, depending on the respective units. Defining the baseline as 1 (X0) and plotting all following time points as delta (X1-X0) would make the graphs quite confusing. Removing the baseline (defined as 1) from the visualization for the sake of clarity, we would deny the reader an important piece of information. So, we struggle to implement your suggestion without adding complexity or truncating the information. We adjusted all measurements for baseline values for the linear mixed model analyses and the visualization with forest plots though. 

Do the authors have data on white blood cell counts or CRP levels after embolization to assess whether systemic inflammation differs between proximal and distal embolization?

Unfortunately, we obtained no data on systemic inflammation as we focused primarily on surrogate parameters of liver function.

The increase in GPT levels at 3 days alongside similar levels at 30 days could indicate transient liver hypoperfusion after the procedure but a similar long-term outcome. Please discuss.

Thank you very much for this helpful remark. We expanded on this in the discussion.

Minor points:

The authors state that there were no procedure-related complications. Does this apply to pain, splenic abscesses, and infections?

No splenic abscess or infection was seen. Periinterventional pain was not documented. Unfortunately, there is no accessible documentation on post-interventional pain on the ward. We added this in the Results section.

Were patients routinely treated with nonsteroidal anti-inflammatory drugs, steroids, or antibiotic prophylaxis?

After OLT all patients received a calcineurin-inhibitor-based immunosuppressive protocol and steroids which are usually tapered within the first month after surgery. NSAR or antibiotics are not part of our standard of care for SA embolization but may have been administered at discretion of the referring department depending on the clinical scenario. We added this in the methods section.

Please provide any other characteristics of the study cohort, including time since transplant and immunosuppression, that might affect transaminase progression.

Thank you very much, we included the time to treatment in table 1 (patient characteristics). Unfortunately, we are not able to restore all information on immunosuppression in detail and consistently. Yet, we believe this could in fact be subject to further scientific evaluation.

Was hepatic artery stenosis excluded in all patients?

One patient underwent stenting of the hepatic artery for a dissection but was not included for statistical analyses. Stenosis was excluded in all other patients. Any post-transplant patient is routinely screened with doppler ultrasound for vascular pathologies on a daily routine for at least one week after surgery and is referred to our department in case of findings consistent with e.g. hepatic artery stenosis.

---

## [Decision Letter · Decision Letter 1]

28 Jan 2022

Splenic artery steal syndrome in patients with orthotopic liver transplant: where to embolize the splenic artery?

PONE-D-21-24840R1

Dear Dr. Fleckenstein,

We’re pleased to inform you that your manuscript has been judged scientifically suitable for publication and will be formally accepted for publication once it meets all outstanding technical requirements.

Kind regards,

Pavel Strnad

Academic Editor

PLOS ONE

Additional Editor Comments (optional):

Reviewers' comments:

Reviewer's Responses to Questions

**Comments to the Author**

1. If the authors have adequately addressed your comments raised in a previous round of review and you feel that this manuscript is now acceptable for publication, you may indicate that here to bypass the “Comments to the Author” section, enter your conflict of interest statement in the “Confidential to Editor” section, and submit your "Accept" recommendation.

Reviewer #2: All comments have been addressed

2. Is the manuscript technically sound, and do the data support the conclusions?

Reviewer #2: Yes

3. Has the statistical analysis been performed appropriately and rigorously? 

Reviewer #2: Yes

4. Have the authors made all data underlying the findings in their manuscript fully available?

Reviewer #2: Yes

5. Is the manuscript presented in an intelligible fashion and written in standard English?

Reviewer #2: Yes

6. Review Comments to the Author

Reviewer #2: The authors answered my questions satisfactorily and addressed my concerns. The work has improved significantly after the revision.

7. PLOS authors have the option to publish the peer review history of their article (what does this mean?). If published, this will include your full peer review and any attached files.

Reviewer #2: No

---

## [Editor Report · Acceptance letter]

15 Feb 2022

PONE-D-21-24840R1 

Splenic artery steal syndrome in patients with orthotopic liver transplant: where to embolize the splenic artery? 

Dear Dr. Fleckenstein:

I'm pleased to inform you that your manuscript has been deemed suitable for publication in PLOS ONE. Congratulations! Your manuscript is now with our production department. 

Kind regards, 

on behalf of

Dr. Pavel Strnad 

Academic Editor

PLOS ONE